# Unravelling the mechanism of neurotensin recognition by neurotensin receptor 1

Kazem Asadollahi [1,2,3], Sunnia Rajput[2], Lazarus Andrew de Zhang[3,4], Ching-Seng Ang[2], Shuai Nie[2], Nicholas A. Williamson [2], Michael D. W. Griffin [1,2], Ross A. D. Bathgate [1,3], Daniel J. Scott [1,3], Thomas R. Weikl [5], Guy N. L. Jameson [2,6] & Paul R. Gooley [1,2] ✉

The conformational ensembles of G protein-coupled receptors (GPCRs) include inactive and active states. Spectroscopy techniques, including NMR, show that agonists, antagonists and other ligands shift the ensemble toward specific states depending on the pharmacological efficacy of the ligand. How receptors recognize ligands and the kinetic mechanism underlying this population shift is poorly understood. Here, we investigate the kinetic mechanism of neurotensin recognition by neurotensin receptor 1 (NTS$_1$) using $^{19}$F-NMR, hydrogen-deuterium exchange mass spectrometry and stopped-flow fluorescence spectroscopy. Our results indicate slow-exchanging conformational heterogeneity on the extracellular surface of ligand-bound NTS$_1$. Numerical analysis of the kinetic data of neurotensin binding to NTS$_1$ shows that ligand recognition follows an induced-fit mechanism, in which conformational changes occur after neurotensin binding. This approach is applicable to other GPCRs to provide insight into the kinetic regulation of ligand recognition by GPCRs.

G protein-coupled receptors (GPCRs) are the largest superfamily of membrane proteins, which transduce signals from a wide range of stimuli across the plasma membrane to activate intracellular effector proteins, most importantly G proteins, GPCR kinases, and the β-arrestin family of proteins[1]. Ligand binding to the extracellular, so-called orthosteric, ligand-binding pocket of GPCRs induces global conformational alterations in the seven transmembrane helix domain (TMD), including the prominent outward movement of helix 6 (TM6) and inward movement of TM7 in the cytosolic region[2]. These conformational changes open the intracellular crevice for the binding and subsequent activation of G proteins[3]. Activation of G protein is followed by desensitization of the receptor largely via GPCR receptor kinase (GRK) phosphorylation-induced β-arrestin-mediated receptor internalization[4].

GPCRs usually exhibit a basal level of activity in the apo-state that is modulated in an efficacy-dependent manner upon ligand binding to the extracellular orthosteric binding site, where agonists fully activate the receptor and inverse agonists suppress the basal activity. The functional output of the receptor can be further modulated by binding of allosteric ligands to secondary binding pockets or by biased ligands that selectively activate one of the signalling pathways over the others[5,6]. Central concepts for understanding such pharmacological behaviour in GPCR signalling are (1) a pre-existing equilibrium of fully inactive to fully active conformations in the conformational landscape of apo-state receptor and (2) a population shift towards the active conformation upon agonist binding[7] that has been supported by NMR, EPR, single molecule studies as well as MD simulations[8–13]. For example, recent NMR studies on Adenosine 2$_A$ (A$_{2A}$) receptor indicate sampling

[1]Department of Biochemistry and Pharmacology, University of Melbourne, Parkville, VIC 3010, Australia. [2]Bio21 Molecular Science and Biotechnology Institute, University of Melbourne, Parkville, VIC 3010, Australia. [3]The Florey, University of Melbourne, Parkville, VIC 3010, Australia. [4]Monash Institute of Pharmaceutical Sciences, Monash University, 381 Royal Parade, Parkville, VIC 3052, Australia. [5]Department of Biomolecular Systems, Max Planck Institute of Colloids and Interfaces, 14476 Potsdam, Germany. [6]School of Chemistry, University of Melbourne, Parkville, VIC 3010, Australia. ✉e-mail: prg@unimelb.edu.au

of active state conformations of the receptor that are further populated in the presence of ligand and heterotrimeric G proteins[8]. The population-shift model addresses the equilibrium aspects of GPCR signalling and allostery, irrespective of the kinetic pathways underlying the coupling of population shift and ligand binding. In the case of two pre-dominant conformations, such as an inactive receptor conformation $R_1$ and an active conformation $R_2$, there are two pathways along which a population shift from $R_1$ to $R_2$ during ligand binding can occur[14] (Supplementary Fig. 1): an induced-fit pathway[15], along which ligand binding to $R_1$ and formation of encounter complexes precede the conformational change to $R_2$ and induce the population shift; and a conformation-selection pathway[16] in which the conformational change from $R_1$ to $R_2$ precedes ligand binding. Despite the pharmaceutical importance of GPCRs, the kinetic mechanism underlying population shifts in GPCRs is poorly understood. Insight into this mechanism may aid tailoring of selective designer molecules with desired pharmacological output against GPCRs.

Neurotensin receptor 1 (NTS$_1$) is a class A GPCR that is primarily expressed in the central nervous system and gastrointestinal tract[17] and activated by the endogenous 13-residue linear peptide neurotensin, pELYENKPRRPYIL[18]. The last six residues of NT (NT8-13) have been demonstrated to be the primary epitope of the peptide for high affinity receptor binding and activation and have been used as a scaffold for development of NTS$_1$ targeting drug candidates[18]. NTS$_1$ regulates neurological processes including dopamine transmission and GABAergic system modulation[19] and is considered as a promising target for treatment of addiction and schizophrenia[20]. Crystal structures of NTS$_1$ in complex with different ligands showed efficacy-dependent modification of the volume of the ligand-binding pocket, where agonists contract the binding pocket and inverse agonists expand its volume[21]. The conformational changes of the binding pocket of NTS$_1$ have been further investigated by NMR[22]. However, how receptor conformational changes and dynamics are kinetically linked to ligand recognition remains unclear.

In this study, we combined $^{19}$F NMR experiments, hydrogen–deuterium exchange mass spectrometry (HDX-MS) and stopped-flow fluorescence kinetics to address the mechanism underlying NT recognition and activation of NTS$_1$. Our ligand-observed $^{19}$F-NMR experiments, on fluorinated full agonist analogues of NT, and receptor-observed experiments, on NTS$_1$ labelled with fluorinated unnatural amino acids, revealed formation of NTS$_1$ conformers upon ligand binding that are in slow conformational exchange. HDX-MS demonstrated that this conformational heterogeneity arises from the interaction between the N-terminal region of the receptor and extracellular loop 2 (ECL2). Further kinetic analysis of binding of NT to NTS$_1$ using stopped-flow fluorescence proposes an induced-fit mechanism of binding underlying NT recognition by NTS$_1$.

## Results

### Development and characterization of fluorinated NT analogues for structural studies

NMR is well-suited to obtain atomic resolution insight into the dynamics of biomolecular systems. However, common isotope labelling schemes, including $^{13}$C and $^{15}$N, are cumbersome to study the dynamics of large systems such as GPCRs. Recently, $^{19}$F-labelled aromatics have re-gained popularity as NMR probes to study conformational dynamics of GPCRs due to the sensitivity of $^{19}$F-aromatics as chemical microenvironmental sensors, resulting in high resolution NMR spectra[23]. For example, fluorinated ligands have been used to unravel the conformational heterogeneity in the orthosteric pocket of the neurokinin receptor[24]. Moreover, the large gyromagnetic ratio of $^{19}$F enables working with low concentrations of the receptor. In this context we were inspired to develop $^{19}$F-NT analogues, by substituting Tyr11 in NT, to investigate the mechanisms of ligand recognition by the receptor[25-28]. We substituted Tyr11 with para-trifluoromethyl-

phenylalanine (tfmF) to produce a sensitive fluorinated analogue of NT (Y11tfmF-NT) (Supplementary Fig. 2) with minimal modification. This substitution reduced the affinity of the peptide for wt-rNTS$_1$ by 100-fold (Supplementary Fig. 3a), which may be due to the steric bulk of the CF$_3$ group and its low tendency to form hydrogen bonds[29]. Perturbing hydrogen bonding of Tyr11 by phenylalanine substitution shows similar effects, but less pronounced than tfmF probably due to the smaller size of phenylalanine compared to tfmF[30]. However, the presence of the CF$_3$ moiety leaves the affinity of Y11tfmF-NT for the engineered NTS$_1$ variant, enNTS$_1$[31], unaffected (Supplementary Fig. 3b), suggesting that the introduced mutations during thermo-stabilization of the receptor (Supplementary Fig. 4) may account for this gain of affinity. Nonetheless, these probes are equally efficacious in BRET assays measuring G protein (Supplementary Fig. 3c) and β-arrestin (Supplementary Fig. 3d) recruitment and can fully activate wt-rNTS$_1$ providing us with a promising tool to investigate the conformational dynamics of ligand binding to the extracellular region of enNTS$_1$.

### The conformational dynamics of ECL2 revealed by $^{19}$F-labelled neurotensin

The synthesized Y11tfmF-NT gives a sharp NMR signal (Supplementary Fig. 5a), that is not perturbed in the presence of DDM (Supplementary Fig. 5b), indicating negligible interaction of the fluorinated peptide with DDM micelles. However, upon complexation of Y11tfmF-NT with enNTS$_1$, two distinct resonances were observed in the $^{19}$F NMR spectrum, a highly populated state of ~75%, and an additional upfield state populated at ~25% (Supplementary Fig. 5c). Both signals are lost upon addition of unlabelled NT validating that both signals correspond to the peptide bound in the orthosteric pocket (Supplementary Fig. 5d); we name them S$_1$ and S$_2$, respectively (Supplementary Fig. 5e). Such signal splitting can be observed in the truncated version of $^{19}$F-NT (Supplementary Fig 5f), Y11tfmF-NT8-13, indicating minimal effect from the first seven residues of NT to the conformation of NT in the bound state. Mutation of Pro10 to alanine, P10A-Y11tfmF-NT8-13, also preserves the conformational heterogeneity (Supplementary Fig. 5g) confirming that the heterogeneity is not due to cis/trans isomerization of Pro10 in Y11tfmF-NT8-13. The signal splitting in the presence of receptor proposes a slow exchange conformational equilibrium in the complex of enNTS$_1$ with Y11tfmF-NT that can be measured using saturation transfer difference (STD) experiments[32]. $^{19}$F STD experiments on Y11tfmF-NT8-13 bound to enNTS$_1$ show that the exchange rate constants from S$_1$ to S$_2$, $k_{12}$, and from S$_2$ to S$_1$, $k_{21}$, are 1.23 s$^{-1}$ and 0.08 s$^{-1}$, respectively (Fig. 1a–c).

Such slow time scale dynamics has been linked to the ligand-independent and large amplitude conformational changes of the extracellular regions of other GPCRs, for example the neurokinin 1 receptor[24]. To support that the source of the conformational dynamics of the receptor is due to fluctuations in the extracellular surface of NTS$_1$, several receptor mutants were made. Two criteria were considered in designing receptor mutants: (1) the mutation should not cause major loss of interactions between the peptide and the receptor based on the crystal structures and (2) the mutation should have minimal effect on the topology of the orthosteric binding pocket. In enNTS$_1$ position 213 is mutated from wild-type arginine to leucine during directed evolution thermostabilization (Supplementary Fig. 4)[31]. Position 213 is not considered as a critical point of direct interaction with bound NT and comparison of the published structures with (PDB: 4XEE)[33] and without (PDB: 4BUO)[34] mutation in position 213 show the topology of the binding pocket is the same. It proposes position 213 as an appropriate candidate for mutation to address the source of conformational heterogeneity. The back mutation of Leu213 to arginine in enNTS$_1$ (enNTS$_1$-R213), resulted in significant stabilization of the S$_1$ conformer, increasing its population from ~25% in enNTS$_1$ to ~60% in enNTS$_1$-R213 (Fig. 1f). However, mutation from a

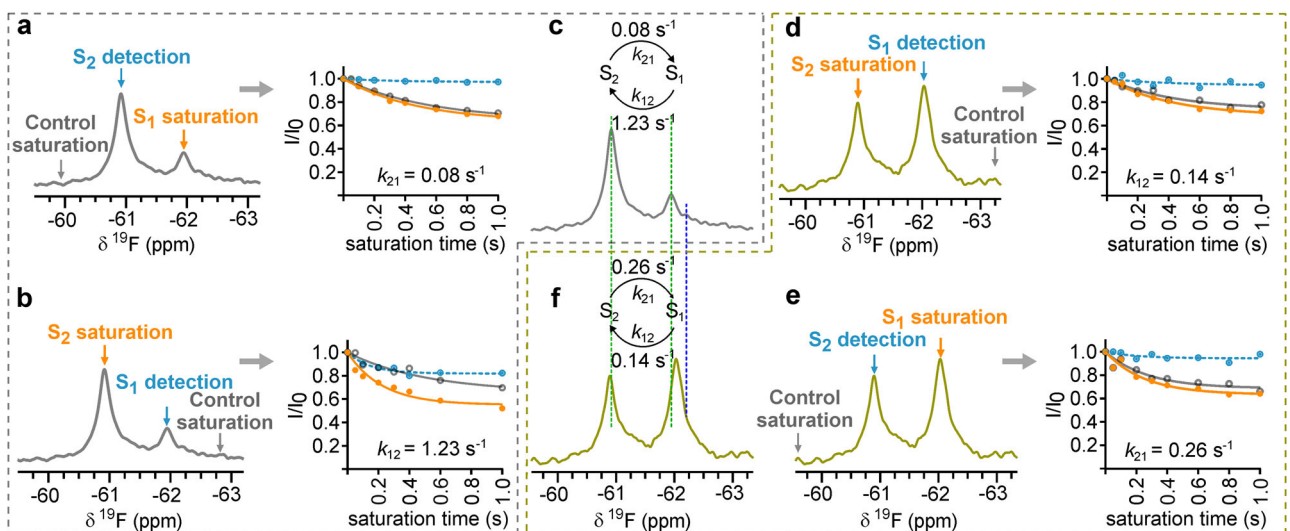

**Fig. 1 | $^{19}$F NMR of Y11tfmF-NT8-13 shows slow-exchanging conformational heterogeneity in complex with enNTS$_1$ and enNTS$_1$-R213.** $^{19}$F STD experiments on **a**, **b** the complex of Y11tfmF-NT8-13 and enNTS$_1$ and **d**, **e** Y11tfmF-NT8-13 and enNTS$_1$-R213. In each experiment the intensity of the detected peak was measured by on-resonance irradiation (orange line, solid symbols) and equidistant off-resonance irradiation (grey line, open symbols) from the detected peak as control. The on- and off-resonance spectra were subtracted to obtain the saturation transfer exchange plots (dashed light blue line, filled symbols). In (**c**, **f**), the exchange rates extracted from the saturation plots are summarized above the spectra of Y11tfmF-NT8-13 in complex with enNTS$_1$ (grey) (**c**) and enNTS$_1$-R213 (olive) (**f**). The chemical shift of S$_1$ and S$_2$ are highlighted with green dotted lines. The dark blue line designates the chemical shift of free peptide. In all experiments, a train of 50 ms gaussian shaped pulses with a field-strength of 50 Hz was used for on and off-resonance saturation. Source data for this figure are provided as a Source data file.

hydrophobic residue, Leu213 in enNTS$_1$, to a positively charged residue did not cause any significant chemical shift perturbations, suggesting that the chemical environment around the probe and the topology of the binding pocket of enNTS$_1$ and enNTS$_1$-R213 are similar (Fig. 1f). As expected, the dynamics between the S$_1$ and S$_2$ conformers are modified, whereby $k_{12}$ is 0.14 s$^{-1}$ and $k_{21}$ is 0.26 s$^{-1}$, reflecting a change to the dynamics of ECL2 (Fig. 1d, e).

G protein binding to GPCRs extends the residence time of ligands in the orthosteric binding pocket via allosterically inducing conformational changes in the extracellular surface of the receptor including receptor lidding[35]. To monitor the effect of G protein binding on the conformational dynamics of the complex of enNTS$_1$ and Y11tfmF-NT8-13 we titrated enNTS$_1$-R213 bound to Y11tfmF-NT8-13 with chimeric Gα$_{iq}$. This chimeric construct binds enNTS$_1$ with an affinity of 1 µM and has been shown to induce small intensity or chemical shift changes to the $^{13}$C$^\varepsilon$H$_3$ resonances of Met-204, which is near the orthosteric binding pocket[36]. However, titration of Y11tfmF-NT8-13 bound to enNTS$_1$-R213 with Gα$_{iq}$ did not induce any further changes to the conformational dynamics of the complex (Supplementary Fig. 6) which may suggest that NT sufficiently stabilizes the active conformational state within the orthosteric region or that as the enNTS$_1$ variant used in this study is thermostabilized and only partially active in cells (Supplementary Fig. 3d, e), Gα$_{iq}$ cannot completely stabilize a fully active state.

## Neurotensin fine-tunes the capping event in enNTS$_1$

The ECL2 of many GPCRs forms a lid over the ligand-bound orthosteric binding pocket of the receptor via interaction with the N-terminal region or ECL3 of the receptor, which can increase the affinity of the ligand by slowing down its dissociation rate[35,37]. Although such a formation has not been confirmed experimentally for NTS$_1$, all structures published to date on the complex of NT with NTS$_1$ show close contacts between the N-terminal region and the ECL2 of NTS$_1$ (PDB: 4XEE, 6YVR, 4BUO)[21,33,34]. However, the structure of these parts of NTS$_1$ in the apo-state or when complexed with small molecule ligands, including full agonist SRI-9829, partial agonist RTI-3a, and the two inverse agonists SR48692 and SR142948A, cannot be resolved, suggesting flexibility of ECL2 and the N-terminal region and that interaction with NT promotes the closed conformation[21]. We hypothesized that the observed conformational dynamics in the complex of fluorinated peptides and the NTS$_1$ variants originates from contact between the receptor N-terminal region and the tip of ECL2 that consequently shrinks the orthosteric binding pocket of the receptor[21]. According to the crystal structures of NTS$_1$ in complex with NT8-13, Pro51 is the first point of contact between the receptor N-terminal region and ECL2, which we designate here as the NECL complex. Regions N-terminal to Pro51 appear of least importance in ligand binding, and are typically truncated in bacterially expressed receptors to minimize heterogeneity through proteolysis[38]. To investigate the interaction of the N-terminal region with ECL2, and how this is influenced by peptide binding, we introduced $^{19}$F labels at Gly50, (Supplementary Fig. 4) close enough to the NECL association point to report on conformational changes in the region and far enough from the NT binding site to minimize the impact of introduced label on ligand binding.

We introduced tfmF at Gly50 of enNTS$_1$, by using the previously engineered tRNA and tRNA synthetase pair for incorporation of tfmF in *Escherichia coli*[39]. This allowed us to produce highly purified, site-specifically labelled receptors for NMR studies (Supplementary Fig. 7). The spectrum of the apo-G50tfmF-enNTS$_1$-R213 showed the presence of two major populations, labelled P1 and P2 (Fig. 2a). Close inspection of P2 showed the presence of downfield asymmetry, whereby deconvolution of P2 suggests an additional population, P3. Mutation of Pro51 in enNTS$_1$-R213 to Ala (G50tfmF-enNTS$_1$-R213-P51A) eliminated the P1 signal but did not change the asymmetry of the peak for P2 and P3, suggesting P1 arises from the cis isomer of Pro51 and P2 and P3 belong to the trans isomer (Supplementary Fig. 8). Titration of NT8-13 into the protein solution promoted sampling of a new substate downfield of P3, labelled P4, to ~90% coupled with line narrowing of the signal (Fig. 2b). The small chemical shift difference between P3 and P4 prevented measurement of the kinetics of exchange between these states by $^{19}$F STD experiments.

If the NECL is formed upon ligand binding and P4 reports this conformational state, repeating the experiments with G50tfmF-enNTS$_1$ (where residue 213 is Leu) should affect the thermodynamics

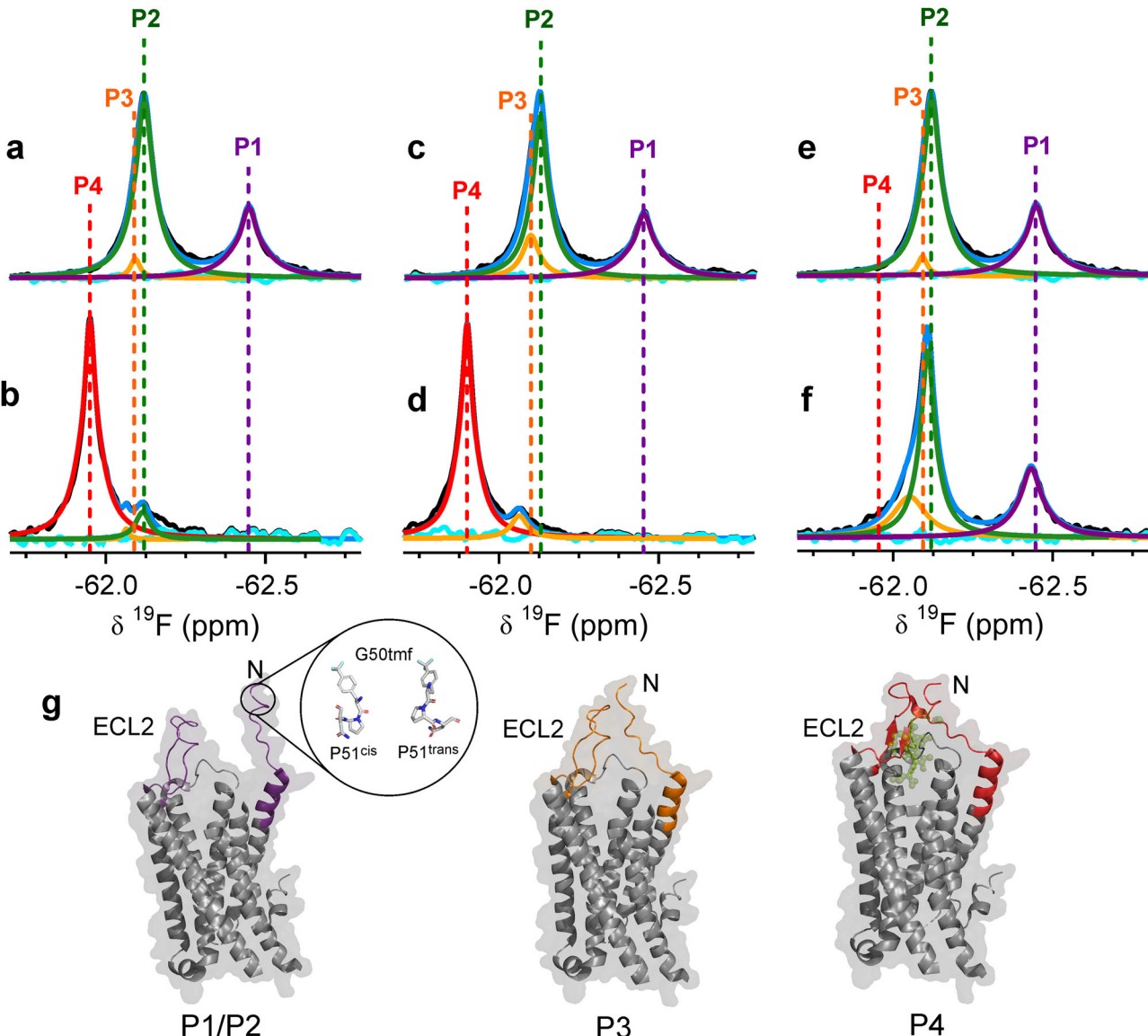

**Fig. 2 | The N-terminal regions of enNTS₁-R213 and enNTS₁ show different conformational dynamics in response to binding agonist and inverse agonist.** $^{19}$F NMR spectra of **a** apo G50tfmF-enNTS₁-R213 and **b** in complex with saturating concentrations of NT8-13. Complexation with NT8-13 promotes formation of P4 (red) and reduces the population of P3 (orange). Despite being saturated, ligand cannot remove P2 (green). P1 (purple), which is due to cis/trans isomerization of Pro51 (Supplementary Fig. 7) disappears. The blue and cyan line in each spectrum indicate the sum and residuals of the deconvoluted spectrum, respectively. $^{19}$F NMR spectra of **c** apo G50tfmF-enNTS₁ (residue 213 is Leu) and **d** in complex with saturating concentrations of NT8-13. Colour scheme is the same as described in (**a, b**). P3 is more populated in apo G50tfmF-enNTS₁ compared to G50tfmF-enNTS₁-R213. Upon binding to NT8-13, P1 disappears and P4 formation is promoted, which

is slightly shifted downfield compared to P4 of G50tfmF-enNTS₁-R213 (**b**). $^{19}$F NMR spectra of **e** apo G50tfmF-enNTS₁-R213 and **f** in complex with inverse agonist SR142948A. Colour scheme is the same as described in (**a, b**). P3 is increased in population in the complex with inverse agonist, but the signal P4 is not observed compared to spectra in the presence of peptide agonist. P1 is also observed, suggesting that the N-terminal region retains flexibility. The chemical shift and population of each state is summarized in Supplementary Table 1. **g** Structural models of enNTS₁ where P1/P2 show significant flexibility of the N-terminal region. P4 shows the N-terminal region and ECL2 form a closed conformation upon binding of NT (green). An intermediate state, P3, is formed by loose interaction of ECL2 with the N-terminal region of the receptor. P3 is populated in both apo state and ligand-bound state of the receptor.

of the system. In $^{19}$F spectra of G50tfmF-enNTS₁ the population of P3 increased from ~4% to ~14% (Fig. 2c and Supplementary Table 1). Also absent in the apo state of G50tfmF-enNTS₁, the new substate P4, is again populated to ~90% upon binding of NT8-13. P4, however, is shifted further downfield compared to the P4 signal in the complex of NT8-13 with G50tfmF-enNTS₁-R213 (Fig. 2b, d and Supplementary Table 1), suggesting a slight difference in conformation of the NECL in the two receptors. Another difference between G50tfmF-enNTS₁-R213 and G50tfmF-enNTS₁ is that under saturating concentrations of NT8-13 (10-fold excess), P2 is present in the ligand bound state of G50tfmF-

enNTS₁-R213 and is absent in G50tfmF-enNTS₁ (Fig. 2b, d). Addition of the inverse agonist SR142948A leaves the $^{19}$F signals for G50tfmF-enNTS₁-R213 largely unaffected, and hence is similar to the apo state (Fig. 2e, f). Most notably, P1 persists indicating that the cis/trans isomerism of Pro51 is not affected and there is no evidence of formation of the substate P4. The signal for P2/3 appears more asymmetric and deconvolution shows an increase in P3 (Fig. 2e, f and Supplementary Table 1). These results are largely consistent with the unresolved NECL in crystal structures of the receptor in the apo state and in complex with small molecule ligands, but NECL is strongly populated in the

presence of bound peptide agonists[21]. Similar to enNTS$_1$-R213 bound to Y11tfmF-NT8-13 (Supplementary Fig. 6), the conformational dynamics of G50tfmF-enNTS$_1$-R213 does not appear affected by titration with the chimera Gα$_{iq}$ under our experimental conditions (Supplementary Fig. 9).

### HDX-MS shows different dynamics in the extracellular surface of enNTS$_1$ when bound to agonist and inverse agonist

Hydrogen–deuterium exchange mass spectrometry (HDX-MS) is a powerful emerging technique to investigate the dynamics of proteins, inferred from deuterium uptake, that has been successfully applied to the characterization of membrane proteins[40,41]. The deuterium uptake rates during HDX experiments correlate with the structural fluctuations of hydrogen-bond opening and closing. Differences between protein states, such as apo- and ligand-bound, directly report on hydrogen bond stability in the ligand binding site or of the conformational dynamics of the protein states. To support our $^{19}$F NMR experiments of the receptor and the assignment of the fluorine signals to different conformational states of the receptor, the conformational dynamics of enNTS$_1$ in apo state and in complex with the agonist NT8-13 or the inverse agonist SR142948A were investigated by HDX-MS (Fig. 3). Over the time course of the experiment (0–100 min) there is limited or no hydrogen–deuterium exchange for the helical regions of the TMD embedded within the DDM micelles. The helices of the TMD also showed lower proteolytic cleavage. Nonetheless, the HDX-MS experiments revealed important insight into the effect of ligand binding on the conformational dynamics of the extracellular region of enNTS$_1$ and how ligand binding promotes these changes. Inspection of the available crystal structures shows that the backbone NH of the first 14 residues (G50 to S63) do not engage in hydrogen bonds upon ligand binding. As expected, peptides from this region are insensitive to exchange in both the apo and ligand bound state (Fig. 3). However, peptides that include the first residues of TM1 (D$^{56}$VNTDIYSKV$^{65}$) show marked differences upon complexation with NT8-13 but not SR142948A (Fig. 3b, e), indicating that agonist, and not inverse agonist, binding increases the stability of this helix, which may be due to reduced dynamics of the N-terminal region, consistent with the downfield shift and lower water-accessibility of the resonance assigned to P4 in the $^{19}$F spectra of G50tfmF-enNTS$_1$ (Fig. 2). Moreover, the absence of such protection in the complex of SR142948A with enNTS$_1$ (Fig. 3e), indicates the peptide agonism-dependency of TM1 dynamics. The extracellular ends of TM2 (P$^{122}$VDYN$^{127}$) and TM7 (F$^{344}$DFHYF$^{349}$) (Fig. 3e) show an increase in protection from deuterium exchange for SR142948A bound receptor. SR142948A might impose such effects by direct interaction with the pocket formed between TM6 and TM7, as reported in crystal structures (PDB: 6Z4Q)[21]. Peptides from ECL2 show an increase in protection in either ligand bound state (Fig. 3a, d), although protection appears slightly higher for agonist NT8-13 bound than inverse agonist SR142948A bound (Fig. 3b, e). For the shortest observed peptide of enNTS$_1$ from this region (F$^{206}$TMGLQNL$^{213}$), the NH of Thr207 forms a hydrogen bond at the tip of TM4, while Gly209 and Gln211 of this peptide form hydrogen bonds within ECL2 itself. The remaining residues of this peptide are not expected to form hydrogen bonds. Consistent with the crystal structures, the adamantyl group of SR142948A forms van der Waals interactions with the side chain of Tyr146, Pro227 and Met208 in the ECL2 as well as Leu234 and Ile238 in TM5 which likely reduce the movement of ECL2.

### Kinetic analysis of binding pathway

The appearance of the P4 resonance upon complex formation with NT8-13 and the conformational heterogeneity of the receptor-bound $^{19}$F-labelled peptide raises the question of whether ligand binding is through an induced-fit or conformational-selection mechanism. To address this question, we kinetically analysed the binding of NT to enNTS$_1$ using stopped-flow fluorescence. We previously showed that upon binding of NT to enNTS$_1$ the intrinsic fluorescence of Trp residues in the receptor increases in a dose-dependent manner, probably due to interaction of side chains of NT with Trp339 in ECL3[42]. Equal volumes of NT and enNTS$_1$ were rapidly mixed together using a fixed concentration of 2 μM of the receptor and varying concentrations of the peptide, and fluorescence change was tracked over time (Fig. 4a). Fitting the kinetic traces revealed a double-exponential trace with two characteristic rates $k_1$ and $k_2$ that depend on the total concentration [L]$_0$ of the peptide ligand. The observed rates $k_1$ and $k_2$, as functions of [L]$_0$, were jointly fitted using the equations described previously[43] and summarized in 'Methods'. Fitting the conformational-selection model without constraints on rate parameters leads to an implausibly large probability of the active conformation R$_2$ in the unbound state, and fits in which this probability is constrained to less than 10% poorly match the data (Fig. 4b). In the induced-fit model, in contrast, the observed rates $k_1$ and $k_2$ can be well fitted with plausible conformational excitation and relaxation rate constants $k_e = 0.01 \pm 0.01\,s^{-1}$ and $k_r = 1.1 \pm 0.5\,s^{-1}$ and with a dissociation rate constant $k_- = 0.6 \pm 0.4\,s^{-1}$ of the bound excited state R$_1$L. Moreover, the conformational exchange rate constants $k_e$ and $k_r$ obtained from the stopped-flow data in the induced-fit model are in good agreement with the exchange rate constants $0.08\,s^{-1}$ and $1.23\,s^{-1}$ measured in the $^{19}$F STD experiments of Y11tfmF-NT8-13 bound to enNTS$_1$ within numerical accuracies (Fig. 1). This agreement of conformational exchange rates obtained from distinct experiments is a rather strong confirmation of induced-fit as the binding mechanism and indicates that the states R$_1$L and R$_2$L of the induced-fit model might correspond to the states S$_1$ and S$_2$ of the NMR experiments. A comparably strong kinetic proof of a binding mechanism based on the agreement of conformational exchange rates deduced from NMR and stopped-flow experiments has been previously reported for the rhodopsin kinase/recoverin system[44].

## Discussion

NMR, EPR and single-molecule fluorescence experiments show the co-existence of several conformational states, ranging from inactive conformer to fully active conformer, in the conformational ensemble of apo-state GPCRs[9,45,46]. Ligands with different efficacies modify the population and dynamics of these conformers in the conformational landscape of GPCRs in favour of specific states. For example, in two-dimensional $^1$H,$^{13}$C NMR experiments on the inverse agonist-bound β$_2$AR, the $^{13}$C$^ε$H$_3$ resonance of Met82 splits into two peaks that shift in an efficacy-dependent manner and collapse into one resonance in the presence of full agonist[47]. Similar observations have been made by NMR experiments on similar labelled residues in β$_1$AR[48], α$_{1A}$AR[49], ACKR3[50], MOR[51], adenosine A$_{2A}$[52,53] and NTS$_1$[22] receptors. Recently, the high sensitivity of $^{19}$F-NMR has enabled resolving lowly populated states as well as the measurement of transition rates between these conformations[9,45,52,54–56].

Here, we monitored the conformational ensemble of thermostabilized variants of NTS$_1$ by combining ligand-observed and receptor-observed $^{19}$F-NMR approaches. The fluorinated analogue of NT, Y11tfmF-NT8-13, samples two slow-exchanging conformational states when bound to enNTS$_1$, likely arising from the conformational dynamics of the extracellular surface of enNTS$_1$ that was further supported by receptor mutants at ECL2 (Fig. 1). Conformational change at the extracellular region of GPCRs upon ligand binding has been shown previously by crystallography[21,57] as well as NMR studies[22,24]. The crystal structures of NTS$_1$ in complex with NT (PDB: 6YVR) and small molecule antagonists (PDB: 6Z4Q) show contraction of the binding pocket in an efficacy-dependent manner[21]. Further, several structures show missing density for the N-terminal region and the ECL2 of the receptor in the apo state (PDB: 6Z66) and in complex with small non-peptide ligands (PDB: 6Z4Q, 6Z4S, 6Z8N, 6ZA8, 6ZIN)[21], proposing conformational crosstalk between these regions that inspired our receptor-observed

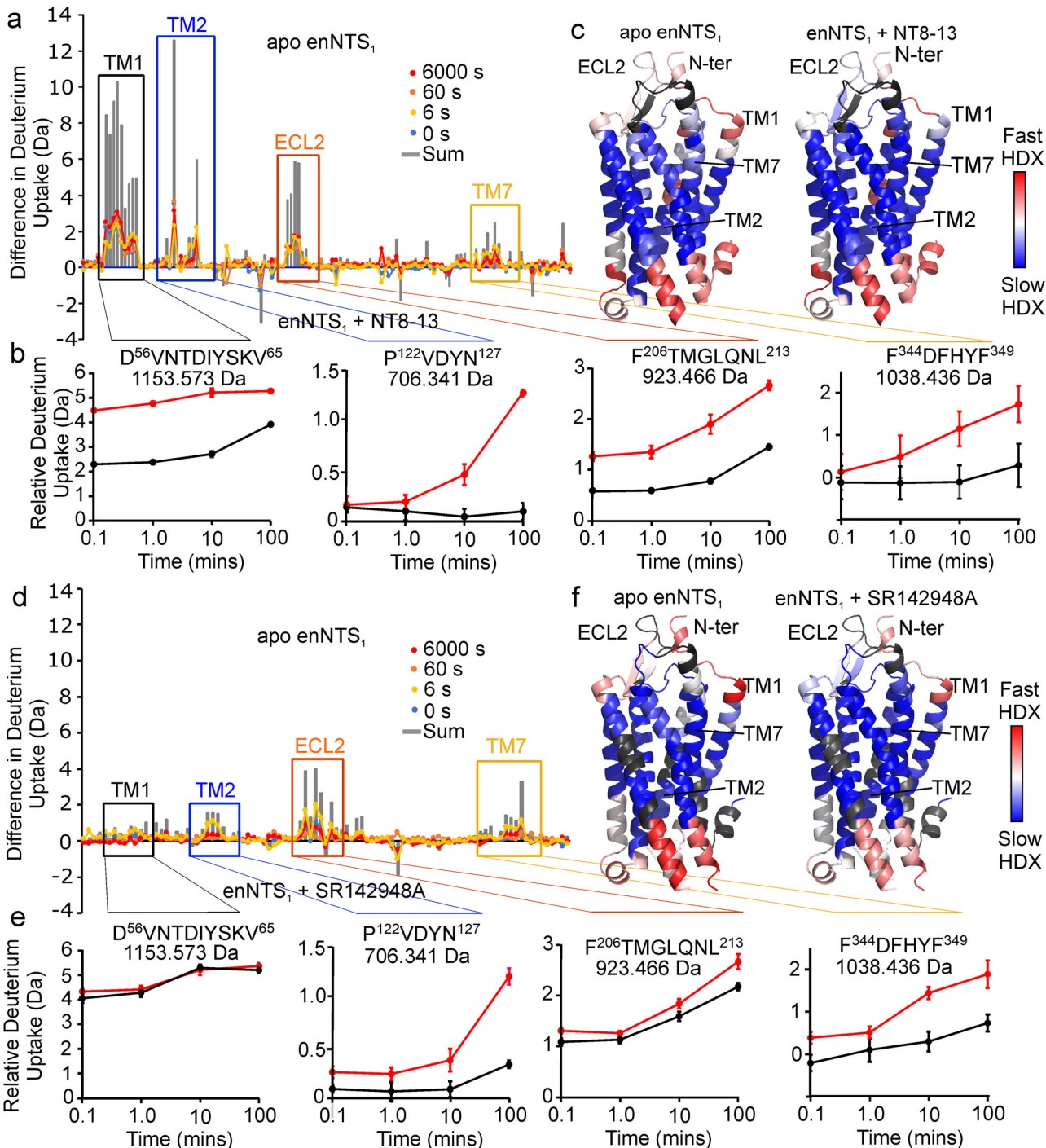

**Fig. 3 | Agonist but not inverse agonist reduces the conformational dynamics of enNTS₁. a** Difference in deuterium uptake plots of enNTS₁ in apo-state and bound to NT8-13 at the time points 0 s (blue); 6 s (yellow); 60 s (orange); 6000 s (red); sum of differences (grey). **b** Relative deuterium uptake plots of apo state (red) and NT8-13 bound (black) for regions that show significant differences in deuterium uptake including the extracellular regions of TM1, $D^{56}VNTDIYSKV^{65}$; TM2, $P^{122}VDVYN^{127}$; TM7, $F^{344}DFHYF^{349}$; and ECL2, $F^{206}TMGLQNL^{213}$. Data shown as intensity-weighted mean values ± standard deviation (n = 4). **c** Heatmap generated from DynamX 3.0 at 100 min overlaid on the structure of NTS₁ (PDB: 4XEE) (blue, least exchange, to red, maximum exchange). Peptides not observed by MS are coloured grey. **d** Difference in deuterium uptake plots of enNTS₁ in the apo-state and bound to inverse agonist SR142948A at time points 0 s (blue); 6 s (yellow); 60 s (orange); 6000 s (red); sum of differences (grey). **e** Relative deuterium uptake plots of apo state (red) and SR142948A-bound (black) show SR142948A has less impact on deuterium uptake than NT8-13 in the extracellular region of TM1 and ECL2 of enNTS₁. Data shown as intensity-weighted mean values ± standard deviation (n = 3). **f** Heatmap generated from HDX data (blue, least exchange, to red, maximum exchange) in apo state and in the presence of SR142948A on the structure of NTS₁ (PDB: 4XEE). Source data for this figure are provided as a Source data file.

experiments. Labelling of the receptor in the N-terminal region at Gly50, G50tfmF-enNTS₁, revealed such spatial proximity between the tip of ECL2 and the N-terminal region of the receptor that is further promoted by the agonist NT8-13 but not the inverse agonist

SR142948A (Fig. 2). The lidding event of GPCRs is a well-established phenomenon for small molecule-bound GPCRs, where binding of ligand promotes folding of ECL2 of the receptor to cover the orthosteric binding pocket thus increasing the residence time of the

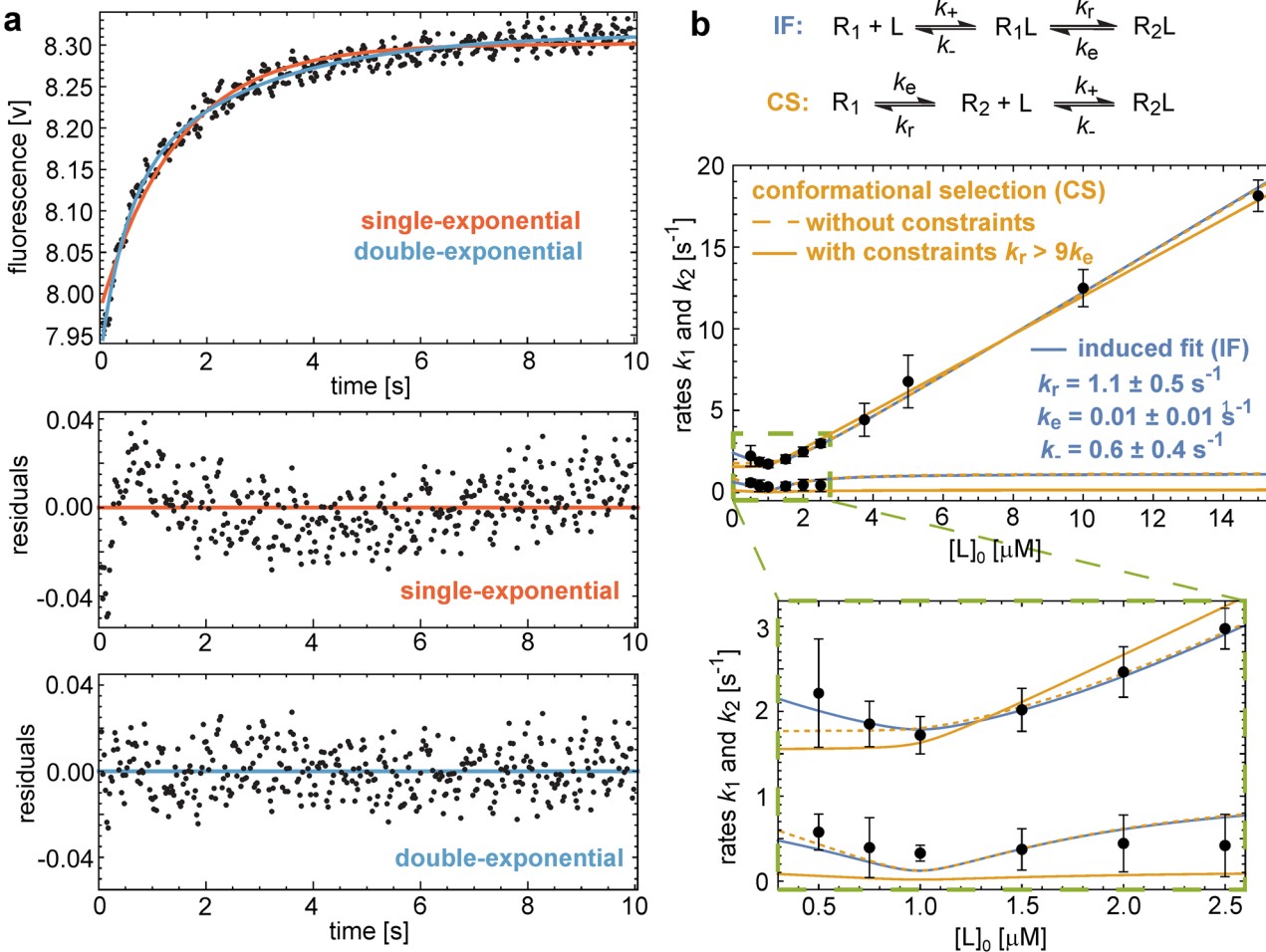

**Fig. 4 | Kinetic analysis of binding of NT to enNTS₁ shows an induced-fit mechanism. a** Single-exponential (red) and double-exponential (blue) fits of an exemplary stopped-flow time trace at 1 μM NT. Fit residuals and goodness of fits assessed with the Akaike Information Criterion (AIC) indicate double-exponential relaxation with two characteristic rates $k_1$ and $k_2$. AIC values −2280.7 and −2454.5 of single- and double-exponential fits, respectively, indicate a significantly better fit for double exponentials. **b** Error-weighted fits of the conformational-selection (CS) (orange lines) and induced-fit (IF) models (blue lines) to the rates $k_1$ and $k_2$ obtained from stopped-flow experiments at different total NT concentrations $[L]_0$ (data points with errors). Data points represent mean values of rates from fits of 12 stopped-flow time traces at 0.5 μM NT; 6 traces at 0.75, 2 and 3.75 μM; 5 traces at 1 and 2.5 μM; 4 traces at 1.5, 10 and 15 μM; 3 traces at 5 μM. Error bars correspond to 95% confidence intervals estimated as 1.96 times standard errors ('Methods'). For

each model, data points are jointly fitted to the model equations for the rates $k_1$ and $k_2$ ('Methods') with the function Nonlinear Model Fit of Mathematica 13[71] using the rate constants $k_e$, $k_r$, and $k_-$ of the CS and IF reaction schemes as fit parameters. The rate constant $k_+$ is replaced by the experimentally determined $K_D$ value of 6 nM (Supplementary Fig. 11). For the CS model, an unconstrained fit (orange dashed lines) with AIC value 15.0 leads to the fit values $k_e = 1.2 \pm 0.5$ s⁻¹, $k_r = 0.6 \pm 0.4$ s⁻¹, and $k_- = 0.005 \pm 0.002$ s⁻¹ and, thus, to an unrealistically large probability of the active state ($R_2$) prior to ligand binding. For more realistic unbound-state probabilities of $R_2$ smaller than 10% imposed by the constraint $k_r > 9 k_e$, the fit of the CS model (full lines) is poor with an AIC value of 33.8 that is significantly larger than the AIC value of 15.1 for the fit of the IF model. Source data for this figure are provided as a Source data file.

ligand[35,57]. We believe our observation reports on similar phenomenon for enNTS₁ where NT binding induces an interaction between ECL2 and the N-terminal region of the receptor and is responsible for the contraction of the binding pocket of NTS₁ upon binding to NT8-13[21], that is further confirmed by HDX-MS experiments on the complex of enNTS₁ with agonist and inverse agonist ligands (Fig. 3).

Although NMR experiments delineate the conformational dynamics of GPCRs and how ligands modify the energy landscape of the receptor, the equilibrium nature of these experiments does not inform on how the population shift is kinetically regulated, which is critical to understand the mechanisms underlying ligand recognition and activation of GPCRs, especially that the output of GPCR function is proposed to be temporally and spatially regulated[58,59]. To address this issue, we successfully applied a label-free stopped-flow fluorescence approach to kinetically study the mechanisms of ligand recognition by enNTS₁. Numerical analysis and mathematical modelling of binding data indicated an induced-fit mechanism underlies ligand recognition

by enNTS₁ including a relatively short-lived intermediate ($R_1L$) state that transitions to the final active ligand-bound state ($R_2L$) (Fig. 4). The physiological importance of such an intermediate state might formulate the kinetic bias that has been reported for other GPCRs, such as the dopamine receptor[60]. Of note, the transition rates calculated from our stopped-flow and ¹⁹F NMR data match the rates of Gs dissociation from β₂AR[61], proposing potential functional relevance of our observed intermediate states.

Whether NT binds to the orthosteric binding pocket of the intermediate state or occupies different conformations in distinct pockets remains to be fully answered. However, using transferred-NOE experiments we recently showed that NT forms an encounter complex with enNTS₁ outside of the orthosteric binding pocket and possibly by engaging the N-terminal region and ECL2[42]. Intermediate-state binding pockets have been attributed to a sequential activation mechanism of the free fatty acid receptor 2, FFA2, by the previously considered allosteric modulator, 4-CMTB[62]. 4-CMTB binds to the orthosteric

binding pocket of FFA2, where it primarily initiates downstream orthosteric signalling followed by diffusion into an allosteric pocket to allosterically modulate receptor activity[62]. The transition of NTS$_1$ from an encounter complex state to the final pose may play a similar role by temporally regulating biased signalling in NTS$_1$. Indeed, impedance-based measurements of NTS$_1$-expressing cell responses to NT and NT8-13 shows significant differences in the response kinetics, albeit at high concentration, where NT produces a more sustained signal compared to NT8-13[63]. Moreover, temporary residence of NT in an intermediate state might also indicate potential secondary binding pockets to be targeted for development of allosteric as well as bitopic ligands that simultaneously bind orthosteric and allosteric sites[64–67]. However, the conformation of NT in the intermediate state remains unresolved.

In this study, we have demonstrated that the binding of the linear peptide NT to NTS$_1$ proceeds via an induced-fit rather than a conformational-selection mechanism. It is possible that peptide-binding GPCRs use induced-fit mechanisms, and the small molecule-binding GPCRs, such as the adrenoreceptors, use conformational-selection mechanisms. However, the slower association rate of ligands to the nanobody-activated or nucleotide free G protein-coupled GPCRs proposes that small molecule ligands may also favour to induce receptor activation through binding to the inactive state of the receptor[35]. The approach we applied here is adaptable to other ligand-GPCR pairs, such as small molecule-binding GPCRs, conditional on the observable change in the intrinsic fluorescence emission upon mixing of ligand with the receptor and providing the availability of NMR probes to investigate the conformational dynamics of the receptor. However, ligands can be conjugated to extrinsic fluorophores for stopped-flow fluorescence assays, in such cases appropriate controls should be conducted to avoid artefacts generated from non-specific interaction of the fluorophore with the membrane environment or receptor surface.

## Methods

### Peptide synthesis and purification
Peptides were synthesized using Fmoc-solid phase peptide synthesis method on a Fmoc-Leu-Wang resin in a 0.025 mmole scale. The peptides were deprotected and cleaved off the resin by using a mixture of 95% TFA, 2.5% water and 2.5% TIPS as a radical scavenging agent. The peptides were further purified using reverse-phase HPLC on a C18 column using a gradient buffer A (0.1% TFA in water) and buffer B (0.1% TFA in acetonitrile). A gradient of 20–30% of buffer B over 20 min was used for purification of these peptides. The peptides were characterized by LC-MS spectrometry on an Exactive Plus Orbitrap mass spectrometer, and the purity was checked (>95%) by analytical HPLC (Supplementary Fig. 2).

### Expression and purification of enNTS$_1$ and enNTS$_1$-R213
The thermostabilized rat neurotensin receptor 1 variant enNTS$_1$-(M208) has been described previously and will be referred to as enNTS$_1$ in this study (Supplementary Fig. 4)[31]. Leu213 of this variant was back-mutated to Arg213 to provide a second receptor model, enNTS$_1$-R213. The plasmids for these receptors were transformed into *Escherichia coli* C43 (DE3) cells and the receptors, N-terminally fused to maltose-binding protein (MBP) and C-terminally with green fluorescent protein (GFP), were expressed in autoinduction media ZYP5052 media supplemented with 1% glycerol, 0.3% lactose and 0.05% glucose[68]. The media was inoculated with 1% overnight preculture and incubated at 37 °C until an OD$_{600}$ of ~1 was reached, then the culture was maintained at 20 °C for another 20–24 h.

All constructs of enNTS$_1$ were purified as follows[31]. The frozen bacterial cell pellet was resuspended in ice-cold lysis buffer (50 mM HEPES pH 7.8, 500 mM NaCl, 20% glycerol, 10 mM imidazole, 4 mM MgSO$_4$, 50 mg lysozyme, 10 mg DNase and one cOmplete protease inhibitor tablet). Two mL of buffer was added per gram of wet weight cells. The cells were stirred in a cold room at 5 °C to achieve complete cell resuspension followed by 8 min sonication on ice with cycles of 10 s on/20 s off. A solution mixture of 4% DDM, 2.4% CHAPS and 0.6% CHS was added to the cell lysate to reach final concentration of 1% DDM, 0.6% CHAPS and 0.12% CHS and lysis was continued for 2 h at 5 °C. The cell lysate was clarified by centrifugation and the supernatant was incubated for 1 h at 5 °C with TALON resin pre-equilibrated with base buffer (25 mM HEPES, 200 mM NaCl, 10% glycerol, 0.05% DDM, 10 mM imidazole). The resin was washed with 100 mL of base buffer containing 500 mM NaCl, 5 mM MgSO$_4$ and 1 mM ATP. The protein was eluted with base buffer containing 200 mM imidazole. Imidazole was removed by buffer exchange and the fusion tags were cleaved off by addition of 1 mM TCEP, 100 mM Na$_2$SO$_4$ and 1 μM 3C-presicion protease to the protein solution and overnight incubation at 5 °C. The cleaved protein was incubated for 1 h with TALON resin pre-equilibrated with cleavage buffer supplemented with 5 mM imidazole and the protein was collected in flow-through. The receptor was concentrated and chromatographed over a Superdex 200 increase 10/300 GL size exclusion chromatography (SEC) column equilibrated in 50 mM phosphate buffer, 100 mM NaCl and 0.05% DDM pH 7.4. The purified protein fractions were pooled together and were either used freshly or snap frozen in liquid nitrogen and stored at −80 °C until required.

### Expression and purification of chimeric Gα$_{iq}$
The chimeric Gα$_{iq}$ was expressed and purified as follows[36]. 4 L of 2YT media, supplemented with 0.2% glucose and 100 μg/mL ampicillin, was inoculated with 1% overnight preculture of *E. coli* BL21 (DE3) cells harbouring the plasmid for expression of chimeric Gα$_{iq}$ N-terminally fused to maltose-binding protein (MBP). The cultures were grown at 37 °C to an OD$_{600}$ of ~0.7. The culture was induced by 1 mM IPTG and expression continued for 16 h at 25 °C. The harvested cells were resuspended in 40 mM HEPES pH 7.5, 100 mM NaCl, 10 mM imidazole, 10% v/v glycerol, 5 mM MgSO$_4$, 50 μM GDP, 10 mg of DNase I, 25 mg lysozyme, and 100 μM DTT by stirring in a cold room at 5 °C to achieve a homogenous suspension. The cells were then lysed by sonication on ice for 16 cycles (10 s on/ 20 s off) and the lysate was clarified by centrifugation at 30,000 × *g*. The supernatant was incubated with Ni-NTA resin pre-equilibrated with 25 mM HEPES, 100 mM NaCl, 10 mM imidazole pH 7.5 for 1 h at 5 °C. The resin was washed with 25 mM HEPES pH 7.5, 500 mM NaCl, 30 mM imidazole, 10% v/v glycerol, 1 mM MgSO$_4$, 50 μM GDP in a gravity flow column and the protein was eluted with 25 mM HEPES, 100 mM NaCl, 300 mM imidazole, 10% v/v glycerol, 1 mM MgSO$_4$, 50 μM GDP, pH 7.5. The imidazole was removed by buffer exchange and the solution was adjusted to 100 mM Na$_2$SO$_4$, 1 mM TCEP and 2 μM home-made 3C-Precision protease followed by overnight incubation at 5 °C. The cleaved protein solution was adjusted to 10 mM imidazole and incubated with Ni-NTA resin pre-equilibrated with 25 mM HEPES, 100 mM NaCl, 10% glycerol pH 7.5. After 1 h incubation at 5 °C, the flow-through was collected as Gα$_{iq}$, concentrated and then chromatographed over a HiLoad 16/600 Superdex 75 pg column equilibrated in 50 mM potassium phosphate pH 7.4, 100 mM NaCl, 10% v/v glycerol, 1 mM MgSO$_4$, 1 μM GDP, 100 μM TCEP. The Gα$_{iq}$ peak fractions were pooled, concentrated, and used either freshly or snap frozen and stored at −80 °C until further use.

### Cell surface competition binding assay
100,000 HEK293F cells (Invitrogen) stably expressing receptor and the cytosolic mCherry fluorophore were suspended in complete phenol red free DMEM (10% FBS, 1% L-Glutamine, 1% penicillin-streptomycin) and were added to the relevant ligand concentrations in the presence of 5 nM Cy5-NT8-13 for enNTS$_1$ and wt-rNTS$_1$ or 2 nM FAM-NT8-13 for rNTS$_1$-R213L. Cells were then gently agitated on an orbital shaker in the

dark for 1 h at room temperature for ligand binding to reach a state of equilibrium. The mean fluorescence intensity of bound FAM-NT8-13 and Cy5-NT8-13 was measured with a CytoflexS (Beckman-Coulter) flow cytometer using the 525/40 and 660/20 filter, respectively. Data points gated for single cells that were mCherry positive as a marker of stable expression. Data was then analysed using the One site – Fit $K_i$ function on GraphPad Prism (version: 9.3.1, Graphpad Software, San Diego, CA). The data for enNTS$_1$ and wt-NTS$_1$ are mean of four replicates ($n = 4$) and the assays with rNTS$_1$-R213L are performed in triplicates ($n = 3$).

## Cell-based assays

500,000 HEK293T cells were seeded per well in a 6-well plate in complete DMEM (10% FBS, 1% L-Glutamine, 1% penicillin-streptomycin). Cells were transfected the following day with a solution of Optimem and lipofectamine 2000 at a quantity of 1 µL/1 µg of DNA. For β-arrestin-2 recruitment assay cells were co-transfected with 250 ng of either pcDNA 3.1 zeo-HA-FLAG-wt-rNTS$_1$-nLuc or pcDNA 3.1 zeo-HA-FLAG-rNTS$_1$-R213L-nLuc and with 600 ng of pcDNA human β-arrestin-2-Venus. In G protein BRET assays, cells were co-transfected with 750 ng of pCSC-rNTS$_1$-IRES-mCherry, pCSC-rNTS$_1$-R213L-IRES-mCherry or pCSC-enNTS$_1$-IRES-mCherry and with 250 ng of Gα$_i$ (for enNTS1) or Gq (for rNTS$_1$ and rNTS1-R213L) BRET biosensor. Cells in each 6-well plate were then washed and harvested with DPBS the following day, then resuspended in 3 mL of complete phenol-red free DMEM (10% FBS, 1% L-Glutamine, 1% penicillin-streptomycin, 25 mM HEPES). Cells were then plated into a white opaque 96-well plate at 80 µL/well. The following day, the media of the cells were changed to 90 µL complete phenol-red free DMEM with 1/500 dilution of Promega Nano-Glo furimazine per well. The plate was then incubated at 37 °C for 10 min in a PHERAstar® FSX microplate reader (BMG LABTECH). Dual luminescence of 450/535 nm was then measured for 3 min before manually adding 10 µL of 10x ligand concentration or vehicle to respective wells before completing the time course measurement. The BRET ratio was calculated as Venus emission to nLuc emission. The following data analysis was performed using GraphPad Prism (version: 9.3.1, GraphPad). The BRET ratio response over time was first normalised to vehicle-treated controls. Concentration–response curves were calculated by first integrating the area under the BRET ratio over time plot using the Area Under the Curve function and then plotted against ligand concentration. Concentration–response curves were then fitted using three-parameter log(agonist) vs response function to calculate Log(EC50). All activity assays are performed in triplicate ($n = 3$).

## Site specific labelling of enNTS$_1$ and enNTS$_1$-R213 with p-trifluoromethyl-phenylalanine

G50 on enNTS$_1$ or enNTS$_1$-R213, was mutated to the TAG amber stop codon and the plasmid was co-transformed into *E. coli* C43 (DE3) cells along with the plasmid containing the genes encoding the engineered tRNA and aminoacyl synthetase pair for incorporation of p-trifluoromethyl-phenylalanine (p-tfmF)[39]. The transformed cells were spread on LB-agar plates supplemented with 100 µg/mL ampicillin and 12.5 µg/mL tetracycline. A single colony was inoculated into an overnight preculture. Expression was conducted in media comprising 50 mM Na$_2$HPO$_4$, 50 mM KH$_2$PO$_4$, 25 mM (NH$_4$)$_2$SO$_4$, 2 mM MgSO$_4$, 1% glycerol, 0.5% Casamino acid (or a mixture of amino acids[39]) and 0.1% glucose. The media was supplemented with 100 µg/mL ampicillin, 12.5 µg/mL tetracycline, a mixture of trace metals[68], 50 mL of 5% Asp solution pH 7.5 per litre of culture and 20 mL of 0.4% Leu solution per litre culture. The culture was started at 37 °C for 1–2 h, then adjusted to 1 mM p-tfmF and the culture continued until an OD$_{600}$ of ~0.6, when protein expression was induced by 0.3 mM IPTG. Expression was continued at 20 °C for another 20–24 h. The protein was purified using the same procedure as for unlabelled enNTS$_1$.

## NMR spectroscopy

The peptide-receptor complex was formed by addition of a 10-fold excess of peptide to receptor. The solution was incubated on ice for one hour, followed by buffer exchange to remove free peptide in 50 mM potassium phosphate pH 7.4, 100 mM NaCl, 0.05% DDM, 50 µM EDTA, 0.02% NaN$_3$. 100 µM of freshly formed complexes were mixed with 10 µM free receptor to ensure excess receptor. The samples were adjusted to 10% D$_2$O, 500 µM DSS and 20 µM TFA as lock nuclei, $^1$H and $^{19}$F references, respectively.

For NMR experiments in the presence of Gα$_{iq}$, enNTS$_1$-R213 was incubated on ice with 5-fold Y11tfmF-NT8-13 over receptor for 30 min and then 5-fold Gα$_{iq}$ over receptor with incubation for another 30 min. Then the buffer was exchanged to G protein buffer, 50 mM potassium phosphate, 100 mM NaCl, 1 mM MgSO$_4$, 100 µM TCEP and 1 µM GDP, pH 7.4, to remove the free peptide. The sample was concentrated and treated with 0.25 IU apyrase for 60 min at room temperature prior to data collection. 10% D$_2$O, 20 µM TFA, 0.05% NaN$_3$ and 500 µM SDS were added to the complex prior to NMR experiments.

30 µM of G50tfmF-enNTS$_1$-R213 was buffer exchanged to G protein buffer and NMR spectra were collected in the presence and absence of 200 µM NT8-13. Another aliquot of the same batch of purified G50tfmF-enNTS$_1$-R213 was buffer exchanged to G protein buffer and incubated on ice with 150 µM Gα$_{iq}$ on ice for 30 min. The sample was then treated with 0.25 IU apyrase followed by incubation for 60 min at ambient temperature prior to data collection. The NMR spectra were then collected in the absence and presence of 200 µM NT8-13. 10% D$_2$O, 20 µM TFA, 0.05% NaN$_3$ and 500 µM SDS were added to the complex prior to NMR experiments.

All NMR experiments were performed on a Bruker 700 MHz Avance IIIHD spectrometer running Topspin 3.2.7 and equipped with a $^1$H/$^{13}$C/$^{15}$N triple-resonance TCI cryoprobe, where the proton channel was tuned to $^{19}$F, -658 MHz. The 1D spectra were collected in a pulse-acquire manner at 298 K, with the carrier frequency set at −65 ppm. 512 to 2048 scans were collected per spectrum with 16 K of data points and a spectral width of 49 ppm. 2680 scans were collected for the sample of Y11tfmF-NT8-13 in complex with enNTS$_1$ in the presence of Gα$_{iq}$. The spectrum on the complex of $^{19}$F-labelled peptides and $^{19}$F-labelled receptor were processed in Topspin vs3.4 using, respectively, 50 Hz and 10 Hz line broadening and zero filling prior to Fourier transformation. Spectral deconvolution was performed in MNova (version: 10.0.2-15465, Mestrelab Research).

$^{19}$F saturation transfer difference (STD) experiments were acquired using the STDDiff pulse sequence from the Bruker library. On-resonance spectra (49 ppm spectral width, 16 K data points and 1.1 s relaxation delay) were collected with 0.05–1 s train of 50 ms gaussian shaped pulses of 50 Hz field strength. These were interleaved with equivalent off-resonance spectra, where the offset was set equidistant to the monitored and saturated peaks. The intensities of the monitored peak in the presence of on-resonance saturation were normalized against the spectra collected in the absence of saturation and subtracted from the intensities in the presence of the equidistant off-resonance spectra to account for off-resonance effects. The obtained intensities plotted against the saturation times were fitted to the Bloch-McConnell equation using GraphPad Prism (version: 9.3.1, GraphPad):

$$I_S(t) = I_S(0) * \left( \frac{k_{SI}}{k_{SI} + R_S} * Exp^{-t(R_S + k_{SI})} + \frac{R_S}{k_{SI} + R_S} \right) \qquad (1)$$

where $I_S(0)$ and $I_S(t)$ are intensities of spin $S$ in the absence and presence of the saturating pulse; $k_{SI}$ and $R_S$ represent the exchange rate constant from spin $S$ to $I$ and the longitudinal relaxation rate for spin $S$.

## Hydrogen−deuterium exchange mass spectrometry

HDX labelling of enNTS$_1$ with and without ligands, NT8-13 and SR142948A, was performed at 20 °C for periods of 0, 6, 60, 6000 s

using a PAL Dual Head HDX Automation manager (Trajan/LEAP) controlled by the ChronosHDX software (Trajan). Purified protein (~25 μM) was incubated on ice with 10-fold excess ligands (~250 μM) before the hydrogen–deuterium exchange reaction to achieve more than ~99% binding site saturation. In the case of NT8-13, which has a high affinity and slow dissociation rate, we diluted out the unbound peptide using Amicon filters down to 1.1-fold over the receptor to minimize potential non-specific interactions. Subsequently, 3 μL of the protein sample was transferred to 57 μL of non-deuterated (50 mM potassium phosphate pH 7.4 containing 100 mM NaCl and 0.02% DDM in H$_2$O) or deuterated (50 mM potassium phosphate buffer pD 7 containing 100 mM NaCl and 0.02% DDM in D$_2$O) buffer and incubated for the respective time. Quenching was performed by adding 50 μL of the deuterated protein to 50 μL of quench buffer (50 mM potassium phosphate buffer, pH 2.3 containing 100 mM NaCl, 0.02% DDM, 200 mM TCEP and 2 M guanidine hydrochloride) at 0 °C. For online pepsin digestion, 80 μL of the quenched sample was passed over an immobilized 2.1 × 30 mm Enzymate BEH pepsin column (Waters) equilibrated in 0.1% formic acid in water (solution A) at 100 μL/min. Proteolyzed peptides were captured and desalted by a C18 trap column (VanGuard BEH; 1.7 μm; 2.1 × 5 mm; (Waters)) and eluted with acetonitrile containing 0.1% formic acid gradient (Solution B) (5% to 40% solution B over 7 min, 40% to 95% solution B over 1 min, 95% solution B for 2 min) at a flow rate of 40 μL/min using an ACQUITY UPLC BEH C18 analytical column (1.7 μm, 1 × 100 mm, (Waters) delivered by ACQUITY UPLC I-Class Binary Solvent Manager (Waters).

For mass spectrometry, a SYNAPT G2-Si mass spectrometer (Waters) was used. Instrument settings were: 3.0 KV capillary and 40 V sampling cone with source and desolvation temperature of 100 and 40 °C, respectively. The desolvation and cone gas flow was at 800 L/h and 100 L/h, respectively. High energy ramp trap collision energy was from 20 to 40 V. All mass spectra were acquired using a 0.4 s scan time with continuous lock mass (Leu-Enk, 556.2771 *m/z*) for mass accuracy correction. Data were acquired in MS$^E$ mode and peptides from non-deuterated samples were identified using Protein Lynx Global Server (PLGS) v3.0 (Waters). To ensure high peptide selection stringency, we applied additional filter constraints of 0.3 fragments per residue, minimum intensity of 2000, maximum MH+ error of 5 ppm, retention time RSD of 10% and peptide has to be identified in at least 70% of the MS$^E$ files. The deuterium uptake values were calculated for each peptide using DynamX 3.0 (Waters). No adjustment was made for deuterium back-exchange during analysis, and therefore all results are reported as relative deuterium exchange levels expressed in mass unit (Da). Deuterium exchange experiments were performed in triplicate for each of the timepoints.

A hybrid significance test[69] consisting of a two-prong statistical test, implemented in Deuteros 2.0[70] was used to identify peptides that show a significant difference (confidence interval of 95%) in deuterium uptake for each pair of HDX-MS experiments (apo versus NT8-13 bound, apo versus SR142948A bound). We also carried out an extra step of curating the data to represent only the peptides that are present in two or more of all replicates (Supplementary Fig. 10).

## Stopped-flow fluorescence spectroscopy

Solutions of different concentrations of NT (1 μM to 30 μM) and a fixed concentration of the receptor (2 μM) were rapidly mixed symmetrically in an Applied Photophysics SX20 stopped-flow spectrophotometer running Pro-Data SX version 2.5 and equipped with a fluorescence detector. Tryptophans in the receptor were excited at 295 nm (slit width 1 mm) and the emitted light with wavelengths longer than 320 nm were collected after passing through a 320 nm cut-off filter. The experiments were performed in 50 mM potassium phosphate, pH 6 containing 100 mM NaCl, 0.05% DDM and the temperature was adjusted to ~10 ± 0.1 °C during experiments using a thermal controller. The slits were adjusted to 0.3 mm. At NT concentrations between 0.75

and 3.75 μM after mixing, 4 to 6 stopped-flow time traces with lengths up to 20 s were recorded. A larger number (12) of such time traces were recorded at 0.5 μM NT. At NT concentrations of 5, 10, and 15 μM, after mixing, 3 to 4 shorter time traces up to 5 s were recorded.

## Fits of stopped-flow time traces

The time traces for NT concentrations up to 3.75 μM were individually fitted with double-exponential fit functions. Mean values of the two fit rates $k_1$ and $k_2$ were obtained by averaging the values from individual fits for the first 10 s of the time traces. For the faster rate $k_1$, the error was determined as the standard error of the individual fit values at a given NT concentration. For the slower rate $k_2$, the error was estimated as the standard deviation of the average values obtained for fits of the time traces up to 5, 10, and 20 s, because the variations of $k_2$ values obtained for these different time windows are larger than the variations between individual trajectories for a given time window. For the NT concentrations ~5 μM and larger, the slower rate $k_2$ cannot be determined reliably from double-exponential fits. Therefore, single-exponential individual fits of the time traces up to 0.5 s were used to determine $k_1$. Fits were conducted with the function Nonlinear Model Fit of Mathematica 13[71] using the method Differential Evolution[72].

## Near-equilibrium relaxation of induced-fit binding

Solving the rate equations of the induced-fit binding model (see Fig. 4b) is complicated by the fact that the binding step is a second-order reaction, which leads to products of the time-dependent concentrations [R$_1$] and [L] of unbound receptors and unbound ligands in the equations. In the standard pseudo-first-order approximation, the rate equations are simplified by assuming that the total ligand concentration [L]$_0$ greatly exceeds the total receptor concentration [R]$_0$, so that the amount of ligand consumed during binding is negligible compared to the total amount of ligand. The concentration of the unbound ligand then can be taken to be constant, and the rate equations only contain terms that are linear in the time-dependent concentration of the receptor, which makes them solvable. In this solution, the time-dependent evolution of the concentrations is a double-exponential relaxation into equilibrium[73,74]. A general solution of the rate equations that holds for all total receptor and ligand concentrations [R]$_0$ and [L]$_0$ can be achieved by expanding the rate equations around the equilibrium concentrations of the bound and unbound receptors and ligands[43]. This expansion leads to a generally valid linearization of the rate equations and captures the final, double-exponential relaxation into equilibrium. The two rates of this double-exponential relaxation process are

$$k_1 = k_e + k_r + \frac{\gamma}{2} + \frac{\sqrt{\gamma^2 + 4k_-k_e}}{2} \tag{2}$$

$$k_2 = k_e + k_r + \frac{\gamma}{2} - \frac{\sqrt{\gamma^2 + 4k_-k_e}}{2} \tag{3}$$

With

$$\gamma = -k_e - k_r + k_- + k_+ \left( \delta - K_D \right) \tag{4}$$

$$\delta = \sqrt{([L]_0 - [P]_0 + K_D)^2 + 4[P]_0 K_D} \tag{5}$$

and with the overall dissociation constant

$$K_D = \frac{k_- k_e}{k_+ (k_e + k_r)} \tag{6}$$

of the induced-fit binding model. This general result for the two rates of the final, double-exponential relaxation includes the result derived

in the pseudo-first approximation as a special case in the limit of large total ligand concentrations $[L]_0$.

## Near-equilibrium relaxation of conformational-selection binding

As in the case of the induced-fit binding model, a general solution of the conformational-selection binding model (see Fig. 4b), which holds for all total receptor concentrations $[R]_0$ and ligand concentrations $[L]_0$, can be achieved by expansion of the rate equations around the equilibrium concentrations of the bound and unbound receptors and ligands[43]. In this general solution, the two rates of the final, double-exponential relaxation into equilibrium are

$$k_1 = k_e + \frac{\alpha}{2} + \frac{\sqrt{\alpha^2 + \beta}}{2} \tag{7}$$

$$k_2 = k_e + \frac{\alpha}{2} - \frac{\sqrt{\alpha^2 + \beta}}{2} \tag{8}$$

With

$$\alpha = k_r - k_e + \frac{k_-\left((2k_e + k_r)\delta + k_r([L]_0 - [P]_0 - K_D)\right)}{2k_e K_D} \tag{9}$$

$$\beta = 2k_r\left(2k_e - k_- - \frac{k_-(\delta - [L]_0 + [P]_0)}{K_D}\right) \tag{10}$$

and $\delta$ as in Eq. (5), and with the overall dissociation constant

$$K_D = \frac{k_-(k_e + k_r)}{k_+ k_e} \tag{11}$$

of the conformational-selection binding model.

## Biolayer interferometry (BLI) experiments

The affinity of NT for enNTS₁ was measured by BLI experiments on an Octet R8 instrument (Sartorius) equipped with streptavidin (SA) decorated biosensors (Octet® SA Biosensors). The biotinylated receptors at the C-terminus avi-tag were immobilized on the surface of a SA-coated sensor. The free SA sites on the sensor were further quenched by immersing the receptor-coated sensor in a solution of 10 μg/mL biocytin for 10 min. The sensor was washed with buffer and NT association was started by immersing the sensor tip into a 96-well plate containing 200 μL of NT solution, (2–48 nM) and the light interference was measured for 600 s. All experiments were performed in 96-well plates at 15 °C in 50 mM phosphate buffer pH 6, 100 mM NaCl and 0.05% DDM. The BLI response, between 200–250 s, to different concentrations of NT was then calculated by the Octet® BLI analysis 12.2.1.3 (Sartorius) and was plotted against the NT concentrations. The data was then fitted in the GraphPad Prism (version: 9.3.1, GraphPad) to obtain the affinities.

## Reporting summary

Further information on research design is available in the Nature Portfolio Reporting Summary linked to this article.

## Data availability

The mass spectrometry proteomics data have been deposited to the ProteomeXchange Consortium via the PRIDE[75] partner repository with the dataset identifier PXD045464. Other data available from the corresponding author on request. The NMR datasets (Figs. 1, 2, S5, S6, S8, S9), the stopped-flow traces (Fig. 4b) and Cell assay data (Fig. S3) are available on Figshare and can be accessed via https://doi.org/10.6084/m9.figshare.24585621. PDB files referenced in this manuscript are available at the Protein Data Bank (https://www.rcsb.org/): 4XEE, 6YVR, 4BUO, 6Z4Q, 6Z66, 6Z4S, 6Z8N, 6ZA8, 6ZIN. Source data are provided with this paper.

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

## Acknowledgements
This work was supported by National Health and Medical Research Council (NHMRC) project Grants 1081844 and 1141034 (to P.R.G., D.J.S. and R.A.D.B.) and a NHMRC Boosting Dementia Research Leadership Fellowship (to D.J.S.). K.A. is recipient of a Melbourne Research Scholarship. We acknowledge the use of the NMR and Mass Spectrometry facilities at the University of Melbourne. The chimeric Gα$_{iq}$ construct used in this study was a generous gift from A/Prof Joshua Ziarek, Northwestern University.

## Author contributions
K.A., P.R.G., D.J.S., M.D.W.G., R.A.D.B., G.N.L.J. and T.W. conceptualized and designed the research; K.A. performed the protein purification, NMR experiments and Octet assays; K.A. and P.R.G. analysed NMR data; K.A. and S.R. synthesized and purified the peptides; K.A. and G.N.J.L. conducted the stopped-flow fluorescence experiments; K.A., G.N.J.L. and T.W. analysed the stopped-flow data; K.A., S.N., N.A.W. and C.S.A. performed and analysed HDX-MS data; L.A.Z. and D.J.S. performed and analysed cell-based assays; P.R.G. supervised the project; K.A., P.R.G. and T.W. wrote the paper; all authors edited the paper.

## Competing interests
The authors declare no competing interests.
