## [Peer Review File · Nature Communications]

Reviewers' Comments:

Reviewer #1:

Remarks to the Author:

The current manuscript analyzed conformational exchange of NT and NTS1 using ¹⁹F-NMR, HDX-MS, and stopped-flow fluorescence spectroscopy. The authors suggest slow conformational exchange and induced-fit mechanism-based ligand recognition. The current study used well-designed experimental tools to study the conformational exchange upon ligand binding to NTS1. However, there are major and minor concerns to be published in Nature Communications. Most importantly, the currently well-accepted model is "conformational selection" rather than "induced fit" model, and thus it is necessary to provide more detailed rationale to support the author's hypothesis (described below at #1 of the major concerns).

Major concerns are below.

1. As the authors mentioned in the manuscript, there are number of published data suggesting that "NMR, EPR and single molecule fluorescence experiments show the co-existence of several conformational states, ranging from inactive conformer to fully active conformer, in the conformational ensemble of apo-state GPCRs", and most of these studies suggest that the conformational changes of GPCRs between inactive and active states follows "conformational selection" but not "induced fit" model. However, the current study suggests "induced fit" model for the NT recognition of NTS1. Would this be due to a unique characteristic of NTS1 or due to analyzing different region of the receptor? Please provide rationale for this discrepancy or perform similar analysis with A2A or b2AR, the well-established model GPCRs.

2. The closing of the extracellular part of the receptor is prominent upon G protein binding (based on ref 35 in the current manuscript). Please further analyze the effect of G protein binding in the closing of the extracellular part of the receptor.

3. It is very hard to understand the HDX-MS data. Please provide residue numbers of each peptide in the uptake plot graphs and indicate the regions on the structure. It would be helpful to use NT-bound structure for Figure 3a-c and SR142948A-bound structure for Figure 3d-f. In the uptake plot the maximum of y-axis is 8 for all the peptides. This may mis-lead the readers. Please use the maximum of y-axis as the maximum uptake for each peptide. Please perform statistical analysis to show the significant differences between apo and ligand-bound state of each peptide. The HDX-MS society provided guideline for reporting HDX-MS data (Masson et al. 2019, DOI: <https://doi.org/10.1038/s41592-019-0459-y>), and thus please provide HDX data summary and HDX data tables according to this guideline. Without these corrections and supplementary data, it is not possible to clearly examine the HDX-MS data.

Minor concerns are below.

1. Check that the subscripts are correctly placed in Figure 2 legend.

2. Make sure consistency in using) or () in Figure 3 legend.

3. In Figure 3, please be more specific for "Difference (Da)" and "Relative uptake (Da)".

4. In Figure 4b, the error bars are large. Can you argue that these results suggest an induced-fit mechanism?

Reviewer #2:

Remarks to the Author:

In this study the authors explore the kinetic and thermodynamic interplay between the peptide ligands and various constructs of the neurotensin receptor 1 (NTS1) using ¹⁹F-NMR, hydrogen-deuterium exchange mass spectrometry and stopped-flow fluorescence spectroscopy. Excitingly, they observe two slowly exchanging ligand signatures which they validate further by stopped-flow fluorescence. Topological responses are studied through HD exchange MS and used to interpret allosteric response to agonist binding in the receptor, using the inverse agonist as a control. The work is creative and useful to the community. The three techniques and their results really help to cross validate ideas regarding induced ligand recognition and response. For example, the argument for a 3 state versus 2 state model in the stopped flow fluorescence might be hard to justify were it not for corroborating kinetics in the ligand exchange.

It also makes sense to me that the interaction of a peptide ligand with the receptor might follow an induced fit model (versus more long-range responses in TM6 and TM7 where long range cooperativity might necessitate induced fit). The authors may want to bring this up in the discussion if they feel so inclined.

- The figures could benefit from being annotated. For example, Figure 2, I suggest overlaying a 7-TM topology map and a labeling site on the receptor, then add the words (or cartoons) of agonist and inverse agonists and point to the closed structures and P4.

- Figure 3. I suggest adding + agonist and plus inverse agonist directly to the figures, adding circles to regions on the protein that differ and adding a color scale with high exchange at the top.

- Presumably with regard to Figure 1, a large excess of Y111fmF-NT8-13 gives rise to a free peak. Is it the case for neurotensin receptors that the peptide is expected to be recognized by a receptor, desolvated or pre-arranged in some binding complex, and then bound to the orthosteric pocket?

- I understand that P1 associated with the receptor signature is a consequence of a proline isomer. However, is it still possible that P1 is a necessary on-pathway conformer that stabilizes a pre-associated complex?

- It would be really interesting to examine the receptor spectra with mini_G_i. I'd be interested to see what happens to both apo and agonist-bound receptors as a way of viewing allostery and the specific recognition events and responses in TM1 plus kinetics.

- There are several areas in the text where the communication could be tightened. For example "In the uptake plots (b) the apo state is orange symbols and dashed-lines and ligand-bound state grey symbols and closed lines." The paper is however, very well written.

Reviewer #3:

Remarks to the Author:

Asadollahi et al "Unravelling the mechanism of neurotensin recognition by neurotensin receptor 1" investigate the mechanism of neurotensin binding to a thermostabilized construct of neurotensin receptor 1, en2NTS1. A combination of ¹⁹F NMR, H/D exchange MS and stopped-flow fluorescence are used to characterise the product of the NT-receptor interaction and to explore whether binding of the activating ligand to the receptor follows an induced fit or conformational selection mechanism. Understanding the early stages of ligand GPCR interactions is very important for a number of reasons, including the development of novel drugs as well as the understanding of receptor bias, selectivity etc. Our current understanding of this area is very patchy and requires the combination of several biophysical techniques to further our insight. By pursuing this question the authors' study is making an important contribution to the field. The choice of methodologies used seems entirely appropriate and the study is generally carefully executed. The authors conclude that binding follows an induced fit mechanism, however, based on current interpretation of the kinetic data presented I am not fully convinced and there seems scope for further improvement. I would be grateful if the authors could address the issues raised below. Following that I would recommend the manuscript for publication in NC.

1) Line 111 to 116 and Figure S3: The authors show ligand binding to a thermostabilised version of, neurotensin receptor 1, en2NTS1 but don't show G protein or b arrestin activation for this construct. Why not? Without such data, it is not clear to what level the receptor can be activated or whether thermostabilization has any detrimental effects on the cytosolic conformational response of the receptor.

2)

Kinetic data fitting:

A better curve fit in the presence of more adjustable parameters is frequently the case. This calls for an improved significance test for evaluation of double-exponential vs single-exponential fit.

This is important in view of the relatively small differences in fitting on which the key message of the paper relies on.

3)

Kinetic data fitting:

The better fit achieved with a double-exponential could reflect the fact that multiple Trp residues contribute to the intrinsic fluorescence signal, rather than allowing a differentiation between induced fit versus conformational selection mechanism. In en2NTS1 I counted six Trp residues. The authors associate the fluorescence change primarily with a change in the local environment of Trp339 in EL3. This is certainly plausible, however, in the absence of a control experiment where Trp339 has been mutated one does not know whether other Trp are also contributing to the observed signal. In case of the latter a double-exponential fit might just be an attempt to compensate for that. Unfortunately, in the absence of functional G protein or b arrestin coupling data (see point 1) we don't know how much 'activation' the en2NTS1 is able to undergo and accordingly by how much the other regions of the receptor that contain Trp residues undergo conformational changes.

4) NMR:

Please adjust the legend to Fig S5 which currently is inaccurate. The presence of two peaks does not automatically imply that they are in exchange. It is the sattrans experiment that reveals this information.

5)

line 253, 254 etc k1 and k2 are rate constants not rates

REVIEWER COMMENTS

Reviewer #1:

Major concerns are below.

1. ...there are number of published data suggesting that “NMR, EPR and single molecule fluorescence experiments show the co-existence of several conformational states, ranging from inactive conformer to fully active conformer, in the conformational ensemble of apo-state GPCRs”, and most of these studies suggest that the conformational changes of GPCRs between inactive and active states follows “conformational selection” but not “induced fit” model. However, the current study suggests “induced fit” model for the NT recognition of NTS1. **Would this be due to a unique characteristic of NTS1 or due to analyzing different region of the receptor? Please provide rational for this discrepancy or perform similar analysis with A2A or b2AR, the well-established model GPCRs.**

Previous studies on GPCRs have not conducted kinetic experiments and therefore can only propose that the population shift that is observed is consistent with the idea of conformational selection. Our study is therefore novel for GPCRs as we have included such kinetic experiments which currently are the only means to distinguish “conformational selection” from “induced fit”. Nevertheless, it is correct that we cannot say this mechanism is general for GPCRs, nor are we proposing this. The small molecule-activated GPCRs (eg adrenoreceptors, adenosine receptors) may indeed use a different mechanism to the peptide-activated GPCRs, especially linear/disordered peptide activated GPCRs such as the neurotensin receptor. To test this possibility we believe those who are studying these receptors should include non-equilibrium experiments to probe the kinetics. We make an edit in discussion to make this point:

lines 353-357, p13 “It is possible that peptide-binding GPCRs use induced fit mechanisms, and the small molecule-binding GPCRs, such as the adrenoreceptors, use conformational selection mechanisms. However, the slower association rate of ligands to the nanobody-activated receptors or nucleotide free G protein-coupled receptor proposes that small molecule ligands may also favor to induce receptor activation through binding to the inactive state of the receptor.³⁵”

2. The closing of the extracellular part of the receptor is prominent upon G protein binding (based on ref 35 in the current manuscript). **Please further analyze the effect of G protein binding in the closing of the extracellular part of the receptor.**

We have conducted these experiments using the chimeric $G\alpha_{iq}$ (ref 68) that was shown to bind to a similar construct when bound to NT8-13 (K_D , 1 μM). We did not observe any additional changes to the resonances of Y11tFmF-NT8-13 (Figure S6) or the G50tFmF-enNTS₁-R213 (Figure S9) in such a ternary complex and therefore cannot draw further conclusions of dynamic/structural differences. In addition to the supplementary figures we have added the following text:

lines 155-165 p7 “G protein binding to GPCRs extends the residence time of ligands in the orthosteric binding pocket via allosterically inducing conformational changes in the extracellular region of the receptor including receptor lidding.³⁵ To monitor the effect of G protein binding we titrated the enNTS₁-R213 bound to Y11tFmF-NT8-13 with chimeric $G\alpha_{iq}$. This chimeric construct binds enNTS₁ with an affinity of 1 μM and has been shown to induce small intensity or chemical shift changes to the ¹³C^εH₃ resonances of Met-204, which is near the orthosteric binding pocket. However, titration of $G\alpha_{iq}$ to the system did not induce any further changes to the conformational dynamics of the complex (Fig. S6) which may suggest that NT sufficiently stabilizes the active conformational state within the

orthosteric region or that as the enNTS₁ variant used in this study is thermostabilized and only partially active in cells, Gα_{iq} cannot completely stabilize a fully active state.”

lines 219-221 p9 “Similar to enNTS₁-R213 bound to Y11tfmF-NT8-13 (Fig. S6), the conformational dynamics of G50tfmF-enNTS₁-R213 does not appear affected by titration with the chimera Gα_{iq} under our experimental conditions (Fig. S9).”

3. It is very hard to understand the HDX-MS data. **(1)** Please provide residue numbers of each peptide in the uptake plot graphs and indicate the regions on the structure. **(2)** It would be helpful to use NT-bound structure for Figure 3a-c and SR142948A-bound structure for Figure 3d-f. **(3)** In the uptake plot the maximum of y-axis is 8 for all the peptides. This may mis-lead the readers. Please use the maximum of y-axis as the maximum uptake for each peptide. **(4)** Please perform statistical analysis to show the significant differences between apo and ligand-bound state of each peptide. **(5)** The HDX-MS society provided guideline for reporting HDX-MS data (Masson et al. 2019, DOI: <https://doi.org/10.1038/s41592-019-0459-y>), and thus please provide HDX data summary and HDX data tables according to this guideline. Without these corrections and supplementary data, it is not possible to clearly examine the HDX-MS data.

Our responses to these points:

- 1. Residue numbers are now on uptake plots.*
- 2. While we would like to show the heat map on such structure, the structure for SR142948A-bound and apo NTS₁ do not resolve the N-terminal region and/or ECL2, which makes sense according to our data which shows it does not stabilize the N-terminal region. Therefore, we have retained the NT-bound to show the regions that are different using the heat map.*
- 3. We have modified the representative up-take plots where y-axis is maximum for each peptide.*
- 4. We did perform a hybrid significance test to identify differentially changed peptides (Hageman et al. (2019) Anal. Chem. 91:8008-8016). This consist of a two-pronged statistical test, implemented in the Deuterio 2.0 software tool. We have now clarified that in the methods and included in supplementary, Figure S10.*
- 5. All HDX data (raw files, PLGS search results, DynamX results files, HDX data summary) have been deposited, stated under “Data Availability: The mass spectrometry proteomics data have been deposited to the ProteomeXchange Consortium via the PRIDE⁷² partner repository with the dataset identifier PXD045464.” Reviewers can view the HDX data by going to the consortium web page (<http://www.proteomexchange.org/>), enter the identifier which will guide them to the login page where they enter: Username: reviewer_pxd045464@ebi.ac.uk Password: 3aDaNGmq*

Minor concerns are below.

1. Check that the subscripts are correctly placed in Figure 2 legend.

corrected

2. Make sure consistency in using) or () in Figure 3 legend.

corrected to ()

3. In Figure 3, please be more specific for “Difference (Da)” and “Relative uptake (Da)”.

We have edited the label to: “Difference in Deuterium Uptake (Da)” and “Relative Deuterium Uptake (Da)”

4. In Figure 4b, the error bars are large. Can you argue that these results suggest an induced-fit mechanism?

We applied a standard significance test to compare two models with different numbers of fit parameters using the Akaike Information Criterion (AIC) values. These values are:

- for the fit of the IF model (blue lines): AIC = 15.13*
- for the unconstrained fit of the CS model (yellow dashed lines): AIC = 14.99*
- for the constrained fit of the CS model (yellow lines): AIC = 33.78*
- The relative likelihood of the constrained fit of the CS model, compared to the IF fit, is $\exp(-9.33) = 0.000089$.*
- The unconstrained fit of CS and the IF fit have essentially the same AIC values — the small difference of 0.14 is not significant.*

In the constrained fit of the CS model, the relative population of the active conformation R2 in the unbound state (responsible for “basal activity”) is constrained to less than 10% (see caption). So, the argument in the manuscript that the unconstrained fit of the CS model is implausible is because of the resulting fit parameters (i.e. the implausibly large relative population of active conformation R2 (>9-fold inactive R1) in the unbound state resulting from these fit parameters) is crucial. All fits are error-weighted fits, so the large error bars mentioned by the referee are taken into account, in the fits, and in the calculation of the AIC values (both with Mathematica).

To make clear that we have applied these tests we have edited the legend of Figure 4 to include:

“The goodness of the fit between single exponential and double exponential fitting was tested by comparing the Akaike Information Criterion (AIC) score of fits, AIC of -2280.65 and -2454.52 was obtained for single and double exponential fits, respectively, indicating a significantly better fit for double exponentials. AIC values were also compared between the IF, constrained CS and unconstrained CS models being 15.13, 33.78 and 14.99, respectively. Although the unconstrained CS and IF models show similar AIC values, the unconstrained CS model results in an implausible large population of active state in the unbound apo state of the receptor. On the other hand, the likelihood of a constrained CS model compared to an IF model is almost zero.”

Reviewer #2:

1. It also makes sense to me that the interaction of a peptide ligand with the receptor might follow an induced fit model (versus more long-range responses in TM6 and TM7 where long range cooperativity might necessitate induced fit). The authors may want to bring this up in the discussion if they feel so inclined.

We appreciate the comment, but we do not want to make further speculation of other “induced” changes.

2. The figures could benefit from being annotated. For example, Figure 2, I suggest overlaying a 7-TM topology map and a labeling site on the receptor, then add the words (or cartoons) of agonist and inverse agonists and point to the closed structures and P4.

Supplementary Figure S2 shows the labelling point. We have added models of P1 to P4 to the main Figure 2 partly following this suggestion.

3. Figure 3. I suggest adding + agonist and plus inverse agonist directly to the figures, adding circles to regions on the protein that differ and adding a color scale with high exchange at the top.

Figure 3a,c,d,e are labelled with the relevant states. A color scale has been added.

4. Presumably with regard to Figure 1, a large excess of Y11tfmF-NT8-13 gives rise to a free peak. Is it the case for neurotensin receptors that the peptide is expected to be recognized by a receptor, desolvated or pre-arranged in some binding complex, and then bound to the orthosteric pocket?

Yes, a large excess does give a free peak, but our sample preparation minimizes this contribution by buffer exchange prior to NMR experiments and having excess receptor. We have recently published work monitoring ligand binding and we do believe that it forms an encounter complex. We have added to the discussion:

lines 336 to 338 p13 "However, using transferred-NOE experiments we recently showed that NT can form an encounter complex with enNTS₁ outside of the orthosteric binding pocket and possibly involving the N-terminal region and ECL2.⁴¹"

5. I understand that P1 associated with the receptor signature is a consequence of a proline isomer. However, is it still possible that P1 is a necessary on-pathway conformer that stabilizes a pre-associated complex?

We agree or speculate that P1 plays a role in recognition or rate-limiting interaction, where P1 needs to convert to P2 (ie cis to trans). However, this is speculative and would prefer to keep our discussion pointed to our conclusion of the mechanism.

5. It would be really interesting to examine the receptor spectra with mini_G_i. I'd be interested to see what happens to both apo and agonist-bound receptors as a way of viewing allostery and the specific recognition events and responses in TM1 plus kinetics.

As asked by reviewer 1 (comment 2) we conducted ¹⁹F NMR experiments with the addition of a chimeric Gα_{iq}, but no changes were observed. We think kinetic experiments will be very complicated with a ternary complex due to association/dissociation of G protein in the course of reaction that impede further analysis and we prefer not to do these.

6. There are several areas in the text where the communication could be tightened. For example " In the uptake plots (b) the apo state is orange symbols and dashed-lines and ligand-bound state grey symbols and closed lines." The paper is however, very well written.

We have re-read and edited to attempt to "tighten".

Reviewer #3:

1) Line 111 to 116 and Figure S3: The authors show ligand binding to a thermostabilised version of, neurotensin receptor 1, en2NTS1 but don't show G protein or b arrestin activation for this construct. Why not? Without such data, it is not clear to what level the receptor can be activated or whether thermostabilization has any detrimental effects on the cytosolic conformational response of the receptor.

Supplementary figure S3 is now complete with the peptides tested for receptor binding, G protein activation and arrestin recruitment.

2) Kinetic data fitting: A better curve fit in the presence of more adjustable parameters is frequently the case. This calls for an improved significance test for evaluation of double-exponential vs single-exponential fit. This is important in view of the relatively small differences in fitting on which the key message of the paper relies on.

We appreciate the question, and our response is as follows. As described with reviewer 1, a standard significance test to compare two models with different numbers of fit parameters is the Akaike Information Criterion (AIC) (see https://en.wikipedia.org/wiki/Akaike_information_criterion). For the exemplary single- and double-exponential fits in our figure, the AIC values are (given by the program Mathematica used for these fits):

- double-exponential fit: AIC = -2454.52

- single-exponential fit: AIC = -2280.65

This difference is large. What matters is the exponential of half the difference (174/2) of the two AIC values (see wikipedia entry). The relative likelihood of the single-exponential fit is $\exp(-87)$, which is of the order of 10 to minus 38 — in other words, 0.

One could also argue that the residuals of the single-exponential fit in this example indicate that there is another phase, from the “noisy curve” one can see in these residuals.

We have included this test in the legend of Figure 4:

The goodness of the fit between single exponential and double exponential fitting was tested by comparing the Akaike Information Criterion (AIC) score of fits, AIC of -2280.65 and -2454.52 was obtained for single and double exponential fits, respectively, indicating a significantly better fit for double exponentials. AIC values were also compared between the IF, constrained CS and unconstrained CS models being 15.13, 33.78 and 14.99, respectively. Although the unconstrained CS and IF models show similar AIC values, the unconstrained CS model results in an implausible large population of active state in the unbound apo state of the receptor. On the other hand, the likelihood of a constrained CS model compared to an IF model is almost zero.

3) Kinetic data fitting: The better fit achieved with a double-exponential could reflect the fact that multiple Trp residues contribute to the intrinsic fluorescence signal, rather than allowing a differentiation between induced fit versus conformational selection mechanism. In en2NTS1 I counted six Trp residues. The authors associate the fluorescence change primarily with a change in the local environment of Trp339 in EL3. This is certainly plausible, however, in the absence of a control experiment where Trp339 has been mutated one does not know whether other Trp are also contributing to the observed signal. In case of the latter a double-exponential fit might just be an

attempt to compensate for that. Unfortunately, in the absence of functional G protein or b arrestin coupling data (see point 1) we don't know how much 'activation' the en2NTS1 is able to undergo and accordingly by how much the other regions of the receptor that contain Trp residues undergo conformational changes.

We agree that several Trp may contribute to the signal. However, a control mutagenesis of Trp339 is likely to lead to complex problems. This residue has been mutated by Deluigi M et al (Science Advances (2021) 7, eabe5504) who showed that this mutation results in significant loss of ligand binding and receptor function in rat NTS₁. Regardless of the number of Trp or which Trp is contributing, the second event likely arises from a conformational change in the complex, around W339 or any other trp residue. Changes in environment of these Trp can be used for distinguishing between binding pathways and we are taking advantage of the fortuitous change of signal. Moreover, distinguishing between conformational selection and induced fit is derived mainly from the first binding event not the second phase of the reaction, due to the small amplitude of second phase that likely originates from the direct interaction of peptide with W339.

4) NMR: Please adjust the legend to Fig S5 which currently is inaccurate. The presence of two peaks does not automatically imply that they are in exchange. It is the sattrans experiment that reveals this information.

corrected, "slow exchanging" is deleted.

5) line 253, 254 etc k1 and k2 are rate constants not rates

corrected

Reviewers' Comments:

Reviewer #1:

Remarks to the Author:

The authors properly addressed my concerns.

Reviewer #2:

Remarks to the Author:

I'm happy with the revisions and responses

Reviewer #3:

Remarks to the Author:

Asadollahi et al (Unravelling the mechanism of neurotensin recognition by neurotensin receptor 1) have provided a revised manuscript that answers my queries to my satisfaction. I am happy to recommend this manuscript for publication in Nature Communications.

All reviewers are satisfied with our revision as summarized by their comments below. We thank the reviewers for their valuable input.

REVIEWERS' COMMENTS

Reviewer #1 (Remarks to the Author):

The authors properly addressed my concerns.

Reviewer #2 (Remarks to the Author):

I'm happy with the revisions and responses

Reviewer #3 (Remarks to the Author):

Asadollahi et al (Unravelling the mechanism of neurotensin recognition by neurotensin receptor 1) have provided a revised manuscript that answers my queries to my satisfaction. I am happy to recommend this manuscript for publication in Nature Communications.